# Spiky: An ImageJ Plugin for Data Analysis of Functional Cardiac and Cardiomyocyte Studies

**DOI:** 10.3390/jimaging8040095

**Published:** 2022-04-01

**Authors:** Côme Pasqualin, François Gannier, Angèle Yu, David Benoist, Ian Findlay, Romain Bordy, Pierre Bredeloux, Véronique Maupoil

**Affiliations:** 1Groupe Physiologie des Cellules Cardiaques et Vasculaires, Université de Tours, EA4245 Transplantation, Immunologie, Inflammation, 37000 Tours, France; francois.gannier@univ-tours.fr (F.G.); angele.yu@univ-tours.fr (A.Y.); ian.findlay@univ-tours.fr (I.F.); romain.bordy@univ-tours.fr (R.B.); pierre.bredeloux@univ-tours.fr (P.B.); veronique.maupoil@univ-tours.fr (V.M.); 2IHU Liryc, Electrophysiology and Heart Modeling Institute, Fondation Bordeaux Université, 33600 Pessac, France; david.benoist@u-bordeaux.fr; 3Universite de Bordeaux, Inserm, Centre de Recherche Cardio-Thoracique, U1045, 33000 Bordeaux, France

**Keywords:** image and video analysis, excitation-contraction coupling, cardiomyocyte calcium dynamics

## Abstract

Introduction and objective: Nowadays, investigations of heart physiology and pathophysiology rely more and more upon image analysis, whether for the detection and characterization of events in single cells or for the mapping of events and their characteristics across an entire tissue. These investigations require extensive skills in image analysis and/or expensive software, and their reproducibility may be a concern. Our objective was to build a robust, reliable and open-source software tool to quantify excitation–contraction related experimental data at multiple scales, from single isolated cells to the whole heart. Methods and results: A free and open-source ImageJ plugin, Spiky, was developed to detect and analyze peaks in experimental data streams. It allows rapid and easy analysis of action potentials, intracellular calcium transient and contraction data from cardiac research experiments. As shown in the provided examples, both classical bi-dimensional data (XT signals) and video data obtained from confocal microscopy and optical mapping experiments (XYT signals) can be analyzed. Spiky was written in ImageJ Macro Language and JAVA, and works under Windows, Mac and Linux operating systems. Conclusion: Spiky provides a complete working interface to process and analyze cardiac physiology research data.

## 1. Introduction

Recording of action potentials, intracellular calcium transients and contraction evoked by electrical stimulation in cardiomyocytes and cardiac tissue are key steps in physiological and pathological studies of the heart. Experimental exploration of the cardiac system now tends to focus on imaging techniques: action potentials are more and more often recorded by optical mapping techniques [1], intracellular calcium by confocal microscopy, and contraction by video microscopy of isolated cardiomyocytes [2,3] or echocardiography of the whole heart [4]. These new methods provide spatially resolved information which require a new analysis paradigm. Added to this are novel optogenetic methods which explore the behavior of small numbers of targeted cells within tissues [5], requiring spatially resolved data acquisition and analysis.

At the same time, the additional spatial information given by imaging techniques leads to an increased complexity of data analysis. Traditional graphics software cannot perform such analysis. The extraction of raw image data and its analysis requires either the expertise of an image analyst with programming skills or several steps of conversion to make it usable by a biologist, which can be particularly time consuming. Hence, the use of modern techniques, such as optical mapping, is often hampered by the absence of specialists in image analysis and programmers in cardiac laboratories. Therefore, it is important to simplify the analysis methods to make them accessible to more laboratories and to improve reproducibility by bridging the gap between experimental results and image analysis and interpretation.

Several programs exist for the analysis of this type of data, but these are not fully open source and may require licensed software (e.g., Matlab^®^). They may also therefore be difficult to adapt to the user’s specific needs. Moreover, the types of image format that can be processed by these programs are often limited. 

Thus, we developed Spiky as an open source working interface dedicated to the analysis of functional data obtained from experiments going from action potentials to contractions, and from single cells to the whole heart. The final purpose of Spiky was to provide a tool which could be integrated to the data acquisition software (ImageJ) and be able to perform the whole pipeline from data acquisition to results interpretation. It avoids data conversion, exportation, importation and use of multiple software which make the analysis time consuming and possibly error prone.

## 2. Materials and Methods

The datasets generated and analyzed during the current study are available from the corresponding authors on request.

### 2.1. Code Availability

Spiky has been written in JAVA and ImageJ Macro Language. The complete source code is available at http://pccv.univ-tours.fr/ImageJ/Spiky, accessed on 4 February 2022 and GitHub at https://github.com/PCCV/Spiky, accessed on 4 February 2022.

### 2.2. Experimental Data from Single Cardiomyocytes and Whole Heart

#### 2.2.1. Animals

All protocols have been approved and all methods have been performed in accordance with the local ethical committee (Comité d’Ethique en Expérimentation Animale Val de Loire no. 19, Tours, France, permit number 2016090711251954). The study was carried out in compliance with the ARRIVE guidelines.

#### 2.2.2. Action Potentials

Action potentials from single cardiomyocytes were recorded within rat left atria with the sharp microelectrode technique, as described previously [6].

#### 2.2.3. Calcium Transients and Calcium Waves in Isolated Myocytes

Isolated cardiomyocytes were obtained from the left ventricle of adult male Wistar rats. Rats were anesthetized with intraperitoneal injection of pentobarbital (60 mg/kg). The heart was rapidly removed and retrogradely perfused through the aorta. The left ventricle cardiomyocyte isolation was achieved with a Langendorff system, as described previously [7].

Calcium transients of isolated ventricular cardiomyocytes were recorded under confocal microscopy, as described previously [8]. Briefly, ventricular cardiomyocytes were loaded with the calcium-sensitive dye Fluo-4 and superfused with a physiological saline solution. Excitation of the dye was performed at 488 nm and emission was collected at 515 ± 10 nm. Calcium transients were evoked by a square wave of electrical stimulation with a pair of platinum electrodes at a frequency of 1 Hz. To record calcium waves, cardiomyocytes were left in a quiescent state until a calcium wave occurred.

#### 2.2.4. Optical Mapping of Action Potentials Data from Whole Heart

Action potentials were optically recorded on isolated perfused heart as described previously [9]. Briefly, the heart was loaded with the voltage-sensitive dye Di-4-ANEPPS and electrically stimulated with a pair of platinum electrodes. The contraction was then abolished by 10 mM of the electromechanical uncoupler 2,3-butanedione monoxime to avoid motion artifacts. The fluorescent dye was excited at 530 nm and emission was collected at 700 ± 50 nm with a Micam Ultima L camera (SciMeasure Analytical Systems) at the sampling frequency of 1 kHz.

## 3. Results

### 3.1. General Principle of the Algorithm

The principle of the Spiky algorithm is to first perform a detection of all “temporal” peaks in the input signal, whether these are positive or negative to the signal baseline. Peaks are recognized as data points that can be either higher or lower than surrounding points by more than a specific value defined by the user; in Spiky, it corresponds to a percentage of the range between the signal maximum and minimum data values. A correctly defined value allows the user to exclude noise and false positives from the analysis.

The signal input is either bidimensional XT (e.g., table or csv files) or tridimensional XYT (e.g., image stacks; Figure 1). Image stacks or movies are considered to be multiple bidimensional (XT) signals, which can be analyzed together. Then, for each peak of the input signal, all requested peak parameters are computed without requiring any operator intervention (see http://pccv.univ-tours.fr/ImageJ/Spiky/, accessed on 4 February 2022, for a description of the methodology of calculation of each parameter).

The output consists of a representation of the previously computed peak parameters. Several representations are proposed by Spiky, which depend on the type of input data and user requests (e.g., a table of single myocyte contraction parameters, colored map of calcium transients amplitude, isochronal activation map, action potential duration plot or map).

A description of how to use Spiky for several data types is available at http://pccv.univ-tours.fr/ImageJ/Spiky/, accessed on 4 February 2022.

### 3.2. Validation of Spiky

All analysis algorithms used have been validated on several different samples whose manual analysis results were compared with the automated analysis outputs. To test the efficiency and the limits of the peak analyzer, a peak simulator has also been implemented within the plugin. This is accessible in the parameters menu. It allows the creation of “realistic” peaks, so as to help the user to validate their own experimental conditions. Moreover, checking the “Make video” button of this module in the peak simulator window allows the creation of a simulated map.

### 3.3. XT Analysis

XT analysis applies to raw experimental data of only two dimensions or parts of XYT files. For example, the data may be action potentials recorded with a microelectrode (Figure 2), electrically evoked calcium transients or contractions of single cardiomyocytes (video microscopy), cardiac tissue strips (constraint gauge) or whole heart (ultrasound; examples are available at http://pccv.univ-tours.fr/ImageJ/Spiky/, accessed on 4 February 2022).

The output table contains a number of lines equal to the number of events (action potentials, calcium transients or contractions) detected in the file and each column corresponds to one parameter of the event. The table can be exploited directly in ImageJ by creating plots where parameters are plotted against each other, or exported as an .xls file for further investigation or display.

### 3.4. XYT Analysis

XYT analysis particularly applies to image stacks and movies with raw data obtained from optical mapping experiments and confocal microscopy. These techniques allow the optical recording of action potentials and evoked calcium transients at multiple scales, from single cells to the whole heart.

Spiky performs the analysis of two types of information from XYT data: first, signal propagation through the tissue, characterized by the temporal shift of the appearance of the signal in all pixels of the image; second, the shape and parameters of each signal in different areas of the preparation. The analysis allows the creation of isochronal maps where each area of the image is colored according to the signal onset delay, or vector maps showing the direction and speed (norm of the vector) of the signal propagation. It can be applied either to a single cardiomyocyte or to the whole heart, as shown for a spontaneous calcium wave in an isolated cardiomyocyte (Figure 3A) and for action potential propagation in a whole heart (Figure 3B). Thus, this analysis outputs a static visual image of the propagation characteristics (speed, direction, isotropy…) of a signal across a preparation.

The XYT analysis can also be used to compare the behavior of different areas of a preparation. The requested parameter of the signal (e.g., duration, decay constant (τ)) can be computed over time for each pixel of the image. Then, a color is assigned to the pixel depending on the parameter value. If several signals are detected over time, the output parameter map is an average of the parameter values computed for each signal. This analysis can be applied to either a single cardiomyocyte or the whole heart, as shown in Figure 4.

### 3.5. Comparison with Other Existing Algorithms

To our knowledge, there is no existing software with a graphic user interface available for the extraction and analysis of bidimensional peaks from image stacks data. Therefore, the comparison will be performed with other optical mapping analysis software. The main differences between Spiky and other three widely used optical mapping analysis software are presented in Table 1.

## 4. Discussion

To our knowledge, Spiky is the first fully open-source software to propose a wide range of analysis of experimental cardiac XT and XYT data, as well as analysis of optical mapping data. Indeed, despite the existence of two programs whose authors provided the software code, they only apply to optical mapping data analysis [9,10]. The code cannot be adapted in order to fit user’s specific needs without a Matlab^®^ license.

Moreover, Spiky does not require data conversion or programming skills to perform a detailed analysis of excitation–contraction parameters obtained with imaging techniques, since ImageJ opens almost all image file types. The software has been especially developed for a translational use at multiple scales, from single cardiac cell to the whole heart, and from fundamental to clinical research. Though Spiky has been developed for cardiac experimental data, its use can be extended to other cell types and research applications which involve periodic phenomena or peak analysis.

Spiky is free and works under ImageJ, which is a widely used image analysis software for Windows, Mac and Linux operating systems. Moreover, analysis can be performed on any data format that can be opened by ImageJ, from a single XT signal to XYT multichannel images or videos. For those with programming skills, the source code of Spiky is open and can be improved or adapted to fit the user needs.

Limitations: Spiky has been designed for a wide range of applications with different types of data. Therefore, the parameters used for the detection of a peak (positive or negative) have to be entered, tested and validated by the user. Spiky does not support a fully blind analysis and its use does not exempt the user from careful interpretation of the output results. For example, the decay constant map of the calcium transient close to the edge of the cell in Figure 4A is impaired by the contraction of the myocyte, and the measurements of action potential duration on the margins of the tissue in Figure 4B are distorted by edge effects.

Finally, to make it applicable to the largest range of applications yet easy to use, some very specialized functions were not implemented. For example, processing of data from panoramic optical mapping systems. To perform such an analysis, software like RHYTHM would be more appropriate **[9]**.

## 5. Conclusions

Spiky improves the pipeline of cardiac data analysis by making it free, faster and easier for both experienced and inexperienced investigators.

## Figures and Tables

**Figure 1 jimaging-08-00095-f001:**
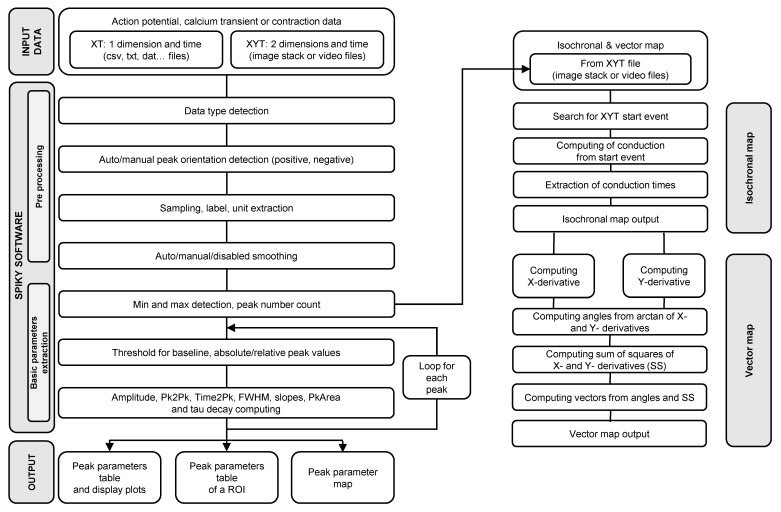
Flow diagram of the Spiky algorithm. Pk2Pk, Time2Pk, FWHM and PkArea correspond respectively to the parameters peak to peak, time to peak, full width at half maximum and area of the peak.

**Figure 2 jimaging-08-00095-f002:**
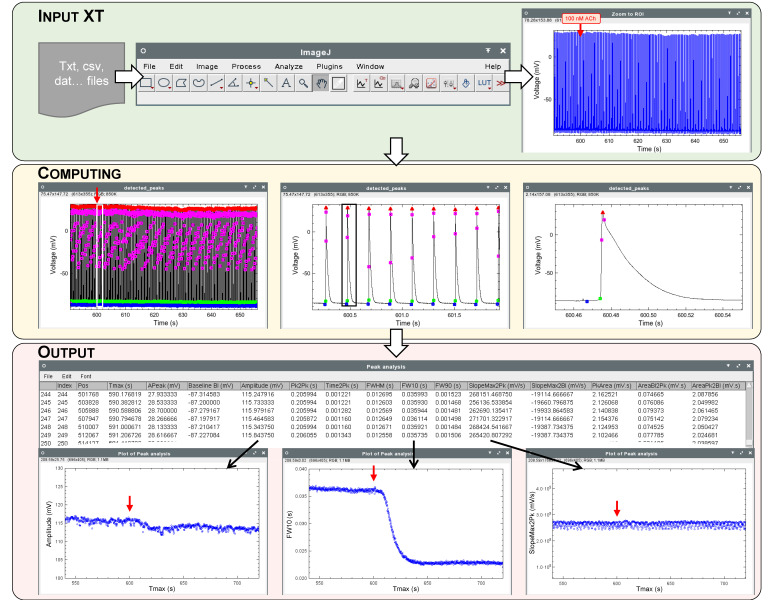
Workflow example of a Spiky XT analysis. The experiment output showed the effect of acetylcholine on rat isolated left atria action potentials recorded with a sharp microelectrode. At 600 s, 100 nM acetylcholine (ACh) was added to the superfusion solution (red arrow). The reduction of the duration of the action potential after acetylcholine superfusion is clearly visible on the second plot obtained from the output analysis values. Symbols on computing plots: red triangle: peak, blue circle: baseline, green circle: threshold, pink circle: max positive/negative slope. Output plots: Left: Amplitude, Center: peak width at 10% of peak amplitude, Right: max slope from baseline to peak.

**Figure 3 jimaging-08-00095-f003:**
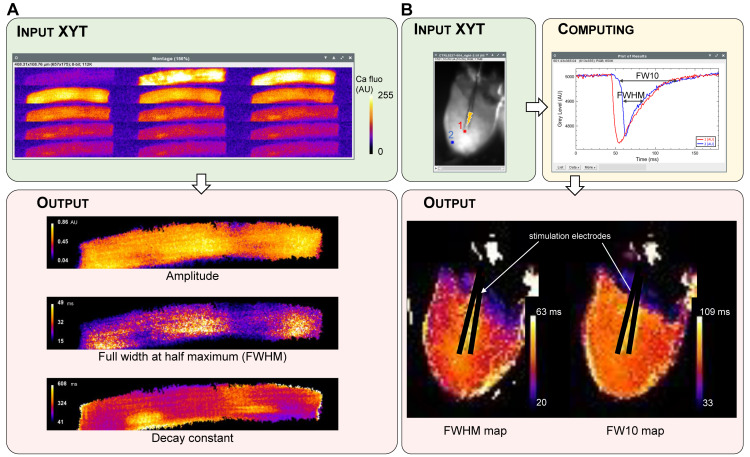
Examples of analysis of XYT data leading to single peak parameter maps as outputs. (**A**) Single rat cardiomyocyte evoked calcium transient recorded with confocal microscopy. The image sequence (left to right and top to bottom) shows a selection of images of the stack with a time interval of 20 ms. Output consists of three maps of relevant parameters of the calcium transient. (**B**) Propagation of an evoked action potential across a rat whole heart. Input image is an image stack obtained from an optical mapping experiment. The position of the stimulation electrodes is indicated by the lightning symbol. The two-colored points illustrate the Spiky software processing occurred for each pixel of the image. After processing, FWHM and FW10 (action potential durations at respectively 50% and 90% of repolarization) are displayed in different areas of the heart as pseudo-colored maps.

**Figure 4 jimaging-08-00095-f004:**
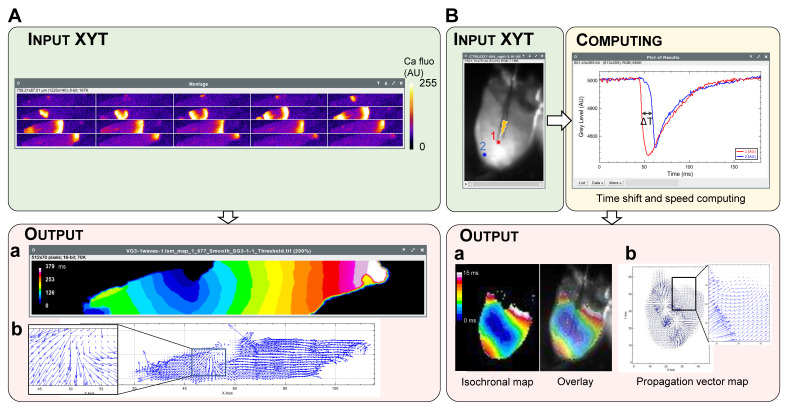
Examples of analysis of XYT data leading to isochronal (**a**) and vector (**b**) maps as outputs. (**A**) Propagation of a spontaneous calcium wave in an isolated cardiomyocyte. Input data is an image stack of the wave recorded by confocal fluorescence microscopy. The image sequence (left to right and top to bottom) shows a selection of images of the stack with a time interval of 20 ms. (**B**) The same heart as in Figure 3B is shown here. The isochronal map (Ba) can be overlaid on the original image.

**Table 1 jimaging-08-00095-t001:** Main characteristics of optical mapping analysis programs.

Software	Platforms	Source Code	Input Data Formats
Spiky	ImageJ	Available (script)	All data or image format types that can be opened with ImageJ and third-party plugins
RHYTHM [9]	MATLAB & Labview	Available but requires licensed software for modifications	TIFF, 4D from 4 cams (XYZT)
ElectroMap [10]	MATLAB	Available but requires licensed software for modifications	TIFF, 3D (XYT)
BV_Ana	Standalone	Unavailable	Micam licensed format

## Data Availability

Not applicable.

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
