# Peer review of "Spiky: An ImageJ Plugin for Data Analysis of Functional Cardiac and Cardiomyocyte Studies"

_2313-433X, 2022, doi:10.3390/jimaging8040095_

Round 1

Reviewer 1 Report

Review:

This manuscript reports an ImageJ plugin, Spiky, to analyze acquired time or image sequence in functional studies of heart tissues or cells. The source code is also available to users. The plugin can analyze two major types of data: the time sequence of action potential and calcium signals acquired by recording systems and the image sequence obtained by optical mapping and microscopy imaging systems. Although similar programs based on Matlab or LabVIEW are available, ImageJ-based programs have many advantages, such as a completely free platform and numerous third-party plugins for image analysis. Considering the ease of use to researchers, this reviewer thinks Spiky is a valuable tool for functional studies in the cardiac research field. However, the quality of writing needs a significant improvement to reach the level of publication. Here list some suggestions for revision.

  • 1 should be included in Methods section.
  • Please provide general algorithms for propagation vector map in Methods section.
  • Please indicate all data formats that can be used in this plugin.
  • Please correct English usage problems. For example, “imagery”, “puzzling”, etc. Many one-sentence paragraphs may not be necessary. For example, Line 86, Line 115, Line 123-129, Line 217, etc.
  • Table 1. “Dependencies” may be replaced by Platforms and “Sources” to “Source Code”.

Author Response

Thank you very much for these useful comments. According to your recommendation, we entirely revised the English language with a native speaker, including editing of one sentence paragraphs and correction of Table 1 titles.

The general algorithm used to obtain isochronal and vector maps is now included in Figure 1.

We have been able to find more than 160 file formats that could be used with this plugin (https://docs.openmicroscopy.org/bio-formats/6.6.1/formats/dataset-table.html). We added in Table 1 a list of those we thought relevant, although not exhaustive, of common image and data formats that can be used as input for Spiky.

Reviewer 2 Report

In this manuscript, Pasqualin et. al. introduced their ImageJ plugin, Spiky, that could conduct heart cell imaging analyses in a more convenient way. The study gives a well-rounded description and introduction of the plugin app. However, there are a couple of questions remain to be answered by the authors.

  1. Personally, I think this manuscript fits better as a method paper than a research article.
  2. In addition to Cardiac cell imaging data, is Spiky also applicable/potentially applicable to other cell types/studies?
  3. Does Spiky support/potentially support 3D volumetric inputs and/or 3D volumetric time-series inputs?
  4. For comparison of Table 1, is there any other plugins available on ImageJ that Spiky can compare with?

Author Response

Thank you very much for your interest in this work and for your very relevant questions.

We totally agree: Spiky manuscript would better fit as a method paper, but this option is not offered by Journal of Imaging, which only accept ‘Research papers’ and ‘Review papers’. That is why we submitted it as a research paper.

In addition to cardiac cell data, Spiky could be used to analyse any periodic events presenting a “peak pattern”. For example: intestinal peristalsis or periodic neuronal activities. We added this information in line 208 to 210 or 237 to 240 in redlined version.

Spiky does not support the analysis of 3D volumetric and 3D volumetric time series. For cardiac 3D volumetric times series, we recommend the use of the software RHYTHM, which has been developed exactly for this type of data (line 226-227 or 260-261 in redlined version).

Despite another extensive research, we did not find any other ImageJ plugin that could be compared with Spiky.

Reviewer 3 Report

  • the authors need to provide the background of the proposed methods
  • compare with existing works 

Author Response

Thank you very much for your comments. According to your recommendation, we entirely revised the English language of the manuscript with a native speaker.

This plugin was initially developed to respond to our specific needs: we needed a tool to analyze a large panel of cardiovascular experimental data, but existing solutions did not suit our needs, in particular as our data could not be opened by other software and proposed analysis did not meet our needs. Using existing software would have required the editing of the source codes, which was impossible because the software were licensed. Thereby, we decided to develop this open source plugin for ImageJ. This allowed the use of image preprocessing tools and other image analysis plugins in the same software (without data conversion/export/import).

This background was added to the introduction (line 48 to 51, or 60 to 63 in redlined version).

We tried to compare Spiky with previous existing works in Table 1. However, as mentioned previously, these other programs do not pursue exactly the same purpose, making the comparison very difficult.

Round 2

Reviewer 1 Report

I don’t think the quality of writing is good enough for a publication. Please revise the English expression. The current version keeps all deleted sentences. It is hard to see the final format. In the next revision, please do not use the “track changes” and only mark the sentences that have been changed.  

In Table 1, it is not necessary to list all formats. Suggest: “All data or image formats that can be opened with ImageJ and third-party plugins.”

Author Response

Thank you for your comments.

We have made some corrections to the text which we hope will increase its comprehension (lines 10, 30-31, 40-44, 133, 203-207).

We have made the corrections to table 1 that you suggested.

We hope that you will find this revision to be acceptable for publication.

Reviewer 3 Report

The major comments have been adequately addressed.

Author Response

Thank you very much for your review.